# *L*^*p*^ Unit Spheres and the *α*-Geometries: Questions and Perspectives

**DOI:** 10.3390/e22121409

**Published:** 2020-12-14

**Authors:** Paolo Gibilisco

**Affiliations:** Department of Economics and Finance, University of Rome “Tor Vergata”, Via Columbia 2, 00133 Rome, Italy; paolo.gibilisco@uniroma2.it

**Keywords:** *L^p^* spheres, *α*-geometries, *α*-Proudman–Johnson equations

## Abstract

In Information Geometry, the unit sphere of $L^{p}$ spaces plays an important role. In this paper, the aim is list a number of open problems, in classical and quantum IG, which are related to $L^{p}$ geometry.


*Gentlemen: there’s lots of room left in $L^{p}$ spaces.*


## 1. Introduction

Chentsov theorem is the fundamental theorem in Information Geometry. After Rao’s remark on the geometric nature of the Fisher Information (in what follows shortly FI), it is Chentsov who showed that on the simplex of the probability vectors, up to scalars, FI is the unique Riemannian geometry, which “contract under noise” (to have an idea of recent developments about this see [1]). So FI appears as the “natural” Riemannian geometry over the manifolds of density vectors, namely over
$$P_{n}^{1}:=\left\{\rho \in R^{n}|\sum_{i}\rho_{i}=1,\;\rho_{i}>0\right\}$$
Since FI is the pull-back of the map
$$\rho \to 2\sqrt{\rho }$$
it is natural to study the geometries induced on the simplex of probability vectors by the embeddings
$$A_{p}\left(\rho \right)=\left\{\begin{array}{cc} p\cdot \rho^{\frac{1}{p}} & p\in [1,+\infty )\\ \log(\rho ) & p=+\infty \end{array}\right.$$
Setting
$$p=\frac{2}{1-\alpha }\;\alpha \in [-1,1]$$
we call the corresponding geometries on the simplex of probability vectors *α-geometries* (first studied by Chentsov himself).

When building similar objects in infinite dimension or in the noncommutative case several interesting questions arise, mostly involving $L^{p}$ spaces.

The purpose of the present paper is to highlight some of the open problems in this area. The epigraph before the Introduction is a half quote of a sentence by Saunders Mac Lane, which is at the beginning of Chapter II in [2]. It is somewhat surprising that Information Geometry suggests some intriguing questions about the geometry of $L^{p}$ spaces.

## 2. The α-Geometries in Finite Dimensions

First of all we may look differently at α-geometries using divergencies. A divergence on an *n*-dimensional manifold *M* is a smooth function
M×M∋(p,q)→D(p,q)∈[0,∞)
which separates points (it is zero iff p=q) such that the matrix
(1)$$g_{ij}^{D}=-\frac{\partial }{\partial p_{i}}\frac{\partial }{\partial q_{i}}{|}_{p=q}D\left(p,q\right)\;i,j=1,...,n$$
defined in a local chart, is strictly positive definite for all p∈M. So any divergence, by the above formula, has an associated Riemannian geometry. Let our manifold *M* be $P_{n}^{1}$ the simplex of strictly positive probability vectors in $R^{n}$ defined in Section 1.

An example of divergence on $P_{n}^{1}$ is the Kullback–Leibler relative entropy, defined as
$$D_{KL}\left(\rho ,\sigma \right):=\sum_{i}\rho_{i}\left(\log\rho_{i}-\log\sigma_{i}\right)$$

Let $\tilde{p}$ be defined by
$$\frac{1}{p}+\frac{1}{\tilde{p}}=1\;\;p\in \left[1,+\infty \right]$$

The α-divergencies are defined as:$$D^{\alpha }\left(\rho ,\sigma \right):=D^{p}\left(\rho ,\sigma \right):=p\cdot \tilde{p}\;\left(1-\sum_{i}\rho_{i}^{\frac{1}{p}}\sigma_{i}^{\frac{1}{\tilde{p}}}\right)\;\alpha \in \left(-1,1\right)$$
$$D^{-1}\left(\sigma ,\rho \right):=D^{1}\left(\rho ,\sigma \right):=D_{KL}\left(\sigma ,\rho \right)$$

The following result is well known.

**Theorem** **1.**
*The geometries generated by the pull-back of the α-embeddings and the geometries generated by the α-divergencies coincide.*


Two complete references for the classical contents of Section 1 and Section 2 can be found in [3,4] while in [5] it is possible to find an overview of the new developments in Information Geometry.

## 3. The Unit Sphere of a (Doubly) Uniformly Convex Banach Space

How to transfer this to infinite dimensions? Let us restrict p∈(1,+∞) and let (X,F,μ) be any measure space. Let *M* be a set of strictly positive probability densities on (X,F,μ), which is endowed with a (possibly infinite dimensional) manifold structure (I remain purposely vague on this point because in moving from finite to infinite dimension a number of delicate analytical questions arise about regularity of the maps involved in these constructions and certainly a comprehensive approach is very much needed).

The function
$$A_{p}\left(\rho \right)=p\cdot \rho^{\frac{1}{p}}\;p\in \left(1,+\infty \right)$$
can be seen as a (smooth) function from *M* to a sphere in the $L^{p}\left(=L^{p}\left(X,F,\mu \right)\right)$ space associated to the above-mentioned measure space. So, what we could pull-back on *M*, say the α-geometry, would be exactly the geometry of the $L^{p}$ sphere.

Following Section 2 in [6] let us show that the sphere of $L^{p}$ space, which is not a Riemann–Hilbert manifold, has some “almost Riemannian” features.

In what follows *X* is a Banach space and $\tilde{X}$ is its dual. We denote by $S^{X}$ the unit sphere of *X* and if $L\in \tilde{X}$ and x∈X we write 〈L,x〉=L(x). If ||x||≤||x+λy||,∀λ∈R we write x⊥y and say that *x* is *orthogonal* to *y*.

The *duality mapping*
$J:X\to P(\tilde{X})$ is defined by
(2)$$J\left(x\right):=\{v\in \tilde{X}:\langle v,x\rangle =||x|{|}^{2}=||v|{|}^{2}\}$$

The space *X* has the *duality map property* if *J* is single valued; in such a case we set $\tilde{x}:=J\left(x\right)$ (by the Hahn–Banach theorem J(x)≠∅ always).

We also say that *X* has the *projection property* if for any closed convex M⊂X and any x∈X there is a unique m∈M such that
(3)||x−m||=inf{||x−z||:z∈M}=d(x,M)
In such a case we set $\pi_{M}\left(x\right):=m$.

**Definition** **1.**
*A Banach space X is doubly uniformly convex (DUC) if X and its dual $\tilde{X}$ are uniformly convex.*


Typical examples of DUC are the $L^{p}$ spaces. In general we have the following properties.

**Proposition** **1.**
*Let X be a DUC Banach space.*
*(i)* 
*X has the projection property.*
*(ii)* 
*X has the duality map property.*
*(iii)* 
*$x⊥\ker(\tilde{x})$.*
*(iv)* 
*If $M:=\ker(\tilde{x})$ then*

$$\pi_{M}\left(v\right)=v-\frac{\langle \tilde{x},v\rangle }{\langle \tilde{x},x\rangle }x$$


**Proposition** **2.**
*Let X be a DUC Banach space.*
*(i)* 
*$S^{X}$ is a Banach submanifold.*
*(ii)* 
*$T_{x}S^{X}$, the tangent space to $S^{X}$ at x, can be identified with ker$\left(\tilde{x}\right)$.*
*(iii)* 
*The projection operator $\pi_{x}:T_{x}X\to T_{x}S^{X}$ is given by*

$$\pi_{x}\left(v\right)=v-\langle \tilde{x},v\rangle$$

*Using this projection, the trivial connection on X induces a connection on $S^{X}$ that we call the natural connection on $S^{X}$.*


When the Banach space *X* is a Hilbert space then the above construction gives nothing else that the Levi–Civita connection on the unit sphere of *X* considered as a Riemann–Hilbert submanifold of *X*. From this it follows that the unit sphere of a DUC Banach space inherits a kind of manageable “Levi-Civita” connection from the trivial geometry of the ambient space.

The above results were proved in [6,7] where they were used to give the first rigorous treatment of α-geometries in infinite dimension. In particular the classical basic formula relating the α-connections to the exponential and mixture connections has been proved:$$\nabla_{p}=\frac{1}{p}\nabla_{m}+\frac{1}{\tilde{p}}\nabla_{e}$$

## 4. Embedding Densities in the Unit Sphere of an Orlicz Space

Beyond the results of Section 3 there is a very simple idea: if we consider a density ρ on an arbitrary measure space (X,F,μ) and Φ is a function on the positive axis, which admits an inverse function $\Phi^{-1}$ then
$$\int \Phi (\Phi^{-1}\left(\rho \right))=\int \rho =1,$$
so that the function
$$\rho \to \Phi^{-1}\left(\rho \right)$$
should embed ρ into the unit sphere of “something”. This very simple (and vague) idea can be made precise by the notion of Orlicz space, which we briefly recall (this was done in [7]).

A *Young function*
Φ is a symmetric convex function Φ:R→R∪+∞ such that Φ(0)=0 and $\lim_{x\to \infty }\Phi \left(x\right)=+\infty$. Let (X,F,μ) be a measure space and f:X→R a measurable function. The *Luxemburg norm*${\left||f|\right|}_{\Phi }$ is given by
$${\left||f|\right|}_{\Phi }:=\inf\left\{r>0:\int_{X}\Phi \left(\frac{\left|f\right|}{r}\right)\leq 1\right\}$$
The *Orlicz space* generated by the Young function Φ is
$$L^{\Phi }:=L^{\Phi }\left(\mu \right):={\left\{f\;\mathrm{is}\;\mathrm{measurable}:||f|\right|}_{\Phi }<+\infty \}$$
If $\Phi \left(x\right)={∥x|}^{p}/p$, p∈[1,+∞) we get the $L^{p}$ space endowed with the equivalent norm ${\left||f|\right|}_{\Phi }=p^{1/p}{\left||f|\right|}_{p}$.

Let us consider now the cases where the Young function Φ is invertible when restricted to the positive axis. If ρ is a density we call $A^{\Phi }\left(\rho \right):=\Phi^{-1}\left(\rho \right)$ the Φ-*embedding*. Trivially we have:$$\int_{X}\Phi \left(\frac{|A^{\Phi }\left(\rho \right)|}{1}\right)=\int_{X}\Phi \left(\Phi^{-1}\left(\rho \right)\right)=\int_{X}\rho =1$$
which implies that $\left|\right|A^{\Phi }\left(\rho \right){\left|\right|}_{\Phi }\leq 1$. Indeed one can prove (see [7]) that
$$\left|\right|A^{\Phi }\left(\rho \right){\left|\right|}_{\Phi }=1,$$
so we may embed any density into the unit sphere of any Orlicz space associated to invertible Young functions.

## 5. Curvature and Scalar Curvature

Let I⊆R be an interval and $\gamma :I\to R^{2}$ a sufficiently regular curve. The *curvature* in the point γ(t):=(x(t),y(t)) is defined as
$$c\left(t\right)=\frac{|x^{\prime }y^{\prime \prime }-x^{\prime \prime }y^{\prime }|}{{\left[{\left(x^{\prime }\right)}^{2}+{\left(y^{\prime }\right)}^{2}\right]}^{3/2}}$$
Curvature coincides with 1/R where *R* is the radius of the osculating circle, namely the circle that gives the best approximation of the curve in a given point.

For a general Riemannian manifold one can introduce the notion of *scalar curvature* according to the following lines. In general if ∇ is an affine linear connection on a manifold *M* the curvature is defined as (see p. 133 in [8])
$$R\left(X,Y\right)Z:=\left[\nabla_{X},\nabla_{Y}\right]Z-\nabla_{\left[X,Y\right]}Z.$$
where X,Y,Z are vector fields. Now consider the case where (M,〈·,·〉) is a Riemannian manifold and ∇ the associated Levi–Civita connection. The Riemannian curvature tensor is defined as (p. 201 in [8])
R(X,Y,Z,W):=〈R(Z,W)Y,X)〉.
Fix now a point ρ∈M and let $\sigma \subset T_{\rho }M$ be a 2-dimensional subspace. Using the exponential map we may associate to σ a 2-dimensional embedded surface $N:=\exp_{\rho }\left(B_{r}\left(0_{\rho }\right)\cap \sigma \right)$ formed by the geodesic segment of length less than *r*, which start tangentially to σ. Let K(σ) denote the Gaussian curvature of *N*. At pages 99–100 in [9] we have the following result.

**Proposition** **3.**
*If u,v is a basis for the plane σ then*
$$K\left(u,v\right):=K\left(\sigma \right)=\frac{R(u,v,u,v)}{\langle u,u\rangle \langle v,v\rangle -{\left|\langle u,v\rangle \right|}^{2}}$$


In particular if $e_{1},e_{2},...,e_{n}$ is an orthonormal basis of $T_{\rho }M$ we have, for i≠j
$$K\left(e_{i},e_{j}\right)=R\left(e_{i},e_{j},e_{i},e_{j}\right)$$
The *scalar curvature* in ρ is defined as
$$\mathrm{Scal}\left(\rho \right):=\sum_{i\neq j}K\left(e_{i},e_{j}\right)$$
From the very definition it is straightforward to deduce that the scalar curvature of an *n*-dimensional sphere of radius *R* is constantly equal to
$$n\left(n-1\right)\cdot \frac{1}{R^{2}}$$

## 6. **Problem 1.** Does the $L^{p}$ Scalar Curvature Behave Like Entropy?

We are ready to discuss the first problem of our list. Let us recall that the α-geometry on $P_{2}^{1}$ is the pull-back geometry induced by the α-embeddings
$$A_{p}\left(\rho \right)=\left\{\begin{array}{cc} p\cdot \rho^{\frac{1}{p}} & p\in [1,+\infty )\\ \log(\rho ) & p=+\infty \end{array}\right.$$
Let $c_{p}\left(\rho \right)$ be the curvature of the α-geometry (where p=2/(1−α)) at the density vector $\rho \in P_{2}^{1}$. One immediately realizes that:

if p=1 then $c_{p}\left(\cdot \right)=\mathrm{constant}=0$;

if p=2 then $c_{p}\left(\cdot \right)=\mathrm{constant}=1/2$.

A straightforward calculation in [10] proves the following result.

**Theorem** **2.**
*If p∈(1,2) then the curvature $c_{p}\left(\cdot \right)$ is a strictly Schur-convex function;*

*If p∈(2,+∞) then the curvature $c_{p}\left(\cdot \right)$ is a strictly Schur-concave function.*


This theorem is “visually” trivial: if you make a picture of the unit spheres of $R^{2}$ endowed with the $L^{p}$ norms you will be convinced of the truth of the statement without any calculation.

It is natural to try to understand what happens in dimension *n*, namely, let us consider the α-embedding on $P_{n}^{1}$ and let Scal${}_{p}\left(\rho \right)$ be the associated scalar curvature. Also in this case one has some trivial cases:

if p=1 then Scalp(·)=constant=0;

if p=2 then ${\mathrm{Scal}}_{p}\left(\cdot \right)=\mathrm{constant}=\frac{1}{4}\left(n-1\right)\left(n-2\right)$.

Indeed for p=1 we have hyperplane and for p=2 we have the geometry of an (n−1)-dimensional sphere whose radius is 2. The following natural conjecture remains open.

**Conjecture** **1.**
*If p∈(1,2) then the scalar curvature Scal${}_{p}\left(\cdot \right)$ is a strictly Schur-convex function;*

*If p∈(2,+∞) then the scalar Scal${}_{p}\left(\cdot \right)$ is a strictly Schur-concave function.*


Some steps toward a proof can be found in [11].

## 7. Petz Theorem

The Chentsov theorem has a noncommutative, “quantum” counterpart, the Petz classification theorem [12,13]. In the quantum case the “noise” is represented by completely positive, trace preserving maps. Let $M_{n}$ be the space of complex n×n matrices, $H_{n}$ the real subspace of Hermitian matrices and $D_{n}^{1}$ the submanifold of (faithful) density matrices, namely
$$D_{n}^{1}:=\left\{\rho \in H_{n}\;|\;\rho >0,\;\mathrm{Tr}\left(\rho \right)=1\;\right\}$$

On the (real) manifold $D_{n}^{1}$ we lose the unicity of the Chentsov theorem: indeed on $D_{n}^{1}$ there are many Riemannian metrics “contracting under noise”. However, Petz was able to characterize all the metrics with this property; these metrics deserve to be called *Quantum Fisher Information(s)*.

**Theorem** **3.**
*There exists a bijective correspondence between Quantum Fisher Information(s) and Kubo–Ando noncommutative means given by the formula*
$${\langle A,B\rangle }_{\rho ,f}:=\mathrm{Tr}\left(A\cdot m_{f}{\left(L_{\rho },R_{\rho }\right)}^{-1}\left(B\right)\right)$$
*where f is the operator monotone function associated to the corresponding mean.*


Obviously $m_{f}{\left(L_{\rho },R_{\rho }\right)}^{-1}$ is a kind of generalized “division by ρ”.

So we have a big family of Riemannian metrics on $D_{n}^{1}$, which play the role of Fisher information in the quantum setting. Among them we are interested in those associated to the following operator monotone functions:$$f_{p}\left(x\right):=\frac{1}{p\cdot \tilde{p}}\cdot \frac{{\left(x-1\right)}^{2}}{\left(x^{1/p}-1\right)\cdot \left(x^{1/\tilde{p}}-1\right)},\;p\in \left(1,+\infty \right)\;\frac{1}{p}+\frac{1}{\tilde{p}}=1$$
$$f_{1}\left(x\right)=f_{\infty }\left(x\right):=\frac{x-1}{\log(x)}\;p=1,+\infty$$
We have that $f_{p}=f_{\tilde{p}}$ and
$$f_{1}=\lim_{p\to 1}f_{p}=\lim_{p\to +\infty }f_{p}=f_{\infty }$$

The quantum Fisher information associated to $f_{1}=f_{\infty }$ is the BKM-metric while the one associated to $f_{p}$ is the WYD(p)-metric.

## 8. **Problem 2.** Geometrization of WYD-Information in Infinite Dimensions?

The WYD(p) metrics are rather special among the quantum Fisher information(s): they are the only one that comes from the pull-back of a dualized pairing, which was proved in [14]. As specified in [15] one can look at this procedure as if we have quantum dynamics associated to a Schrödinger equation, which is embedded using two conjugated α-embeddings. The final result of this procedure is exactly the WYD(p) metric. In particular, for p=2 one sees that the Wigner–Yanase information has a geometric origin, it arises from the pull-back of the map
$$\rho \to 2\sqrt{\rho }$$
as the classical Fisher information [16].

Since WYD information appears in infinite dimensions [17] (Von Neumann algebra setting), it is natural to ask if also in that case one can trace a geometric origin for that object. The ingredients of the previous approach are quantum dynamics, $L^{p}$ spaces and α-embedding: all these objects make sense also in the von Neuman algebra setting; therefore, there is no clear obstacle in this direction.

## 9. **Problem 3.** Petz Conjecture for the BKM Scalar Curvature: A Solution by $L^{p}$ Geometry?

It has been suggested that the scalar curvature of Fisher Information could have a relevant physical meaning in statistical mechanics being linked to the free energy. Maybe stimulated by Petz began the study of the scalar curvature in quantum setting with special emphasis on the BKM metric. He formulated the following conjecture in [18].

**Conjecture** **2.**
*The scalar curvature of the BKM metric is a Schur concave function.*


The truth of the Petz conjecture would be a consequence of the following conjecture.

**Conjecture** **3.**
*There exists ε>0 such that for p∈(1,1+ε) the scalar curvature of the WYD(p) metric is a Schur-concave function.*


Indeed this second conjecture looks much easier to understand than the Petz conjecture. Consider the noncommutative α-geometry on $D_{n}^{1}$ namely the pull-back geometry induced by the α-embeddings
$$A_{p}\left(\rho \right)=\left\{\begin{array}{cc} p\cdot \rho^{\frac{1}{p}} & p\in [1,+\infty )\\ \log(\rho ) & p=+\infty \end{array}\right.$$
exactly as in the commutative case.

Let ${\mathrm{Scal}}_{p}\left(\rho \right)$ be the scalar curvature of the α-geometry (where p=2/(1−α)) at the density matrix $\rho \in D_{n}^{1}$. One immediately realizes that:

if p=1 then Scalp(·)=constant=0;

if p=2 then ${\mathrm{Scal}}_{p}\left(\cdot \right)=\mathrm{constant}=\frac{1}{4}\left(n^{2}-1\right)\left(n^{2}-2\right)$.

Indeed for p=1 we have a hyperplane and for p=2 we have the geometry of a real $\left(n^{2}-1\right)$-dimensional sphere whose radius is 2. Imitating the commutative case we formulate another conjecture.

**Conjecture** **4.**
*If p∈(1,2) then the scalar curvature Scal${}_{p}\left(\cdot \right)$ is a strictly Schur-convex function;*

*If p∈(2,+∞] then the scalar curvature Scal${}_{p}\left(\cdot \right)$ is a strictly Schur-concave function.*


Therefore Conjecture 3 appears rather reasonable: the WYD(p) metric comes from a pair of α-embeddings in duality, from a pair $p,\tilde{p}$ where $1/p+1/\tilde{p}=1$. On the other hand, the BKM metric appears in the limit $p\to 1,\tilde{p}\to +\infty$. For p=1 we have a flat geometry, scalar curvature is zero, and for $\tilde{p}=+\infty$ we see a Schur-concave scalar curvature whose contribution could imply that the BKM scalar curvature has a similar behavior, therefore proving Petz conjecture.

## 10. **Problem 4.** The Exponential Manifold by Orlicz Embedding?

Using the Orlicz spaces (in particular the Zygmund ones) in [19] a Banach manifold structure, called the exponential statistical manifold, has been defined for the space of the strictly positive density functions on an arbitrary measure space.

Because of the existence of the Φ-embeddings of Section 4 it is possible to ask: can the exponential statistical manifold structure be derived (like the α-geometries) from the pull-back of an Orlicz embedding?

## 11. The α-Proudman–Johnson Equations and the α-Connections: The Lenells–Misiolek Result. **Problems 5, 6, 7**

In Problem 1981–29 in the Arnold’s Problems the author asks to find equations of mathematical physics that can be realized as geodesic flows on infinite-dimensional ellipsoids (see page 354 in [20]). This question is natural in the light of the geometric approach to hydrodynamics due to Arnold himself in [21]. In recent years this point of view has led to many similar results, a good reference for this is the Introduction of [20]. Still in recent years there has been a lot of interest in the study of the α-Proudman–Johnson equations, see [22,23,24] for more details. A surprising link between α-geometries and the α-Proudman–Johnson equations has been found by Lenells and Misiolek in [25]. A very rough description is the following.

Let $S^{1}=R/Z$ be the circle, $D(S^{1})$ the group of smooth diffeomorphisms and Rot$\left(S^{1}\right)$ (isomorphic to $S^{1}$) the space of rigid rotations. Using the proper analog of the α-divergences the authors build the α-geometries, and the associated α-connections $\nabla^{\alpha }$ on $D(S^{1})$/Rot$\left(S^{1}\right)$.

Lenells and Misiolek prove in [25] the following result.

**Theorem** **4.**
*The geodesic equation of $\nabla^{\alpha }$ on $D(S^{1})$/Rot$\left(S^{1}\right)$ is the α-Proudman–Johnson equation*
$$u_{txx}+\left(2-\alpha \right)u_{x}u_{xx}+uu_{xxx}=0$$
*In particular, α=0 yields the completely integrable Hunter–Saxton equation*
$$u_{txx}+2u_{x}u_{xx}+uu_{xxx}=0$$
*and α=−1 yields the completely integrable μ-Burgers equation*
$$u_{txx}+3u_{x}u_{xx}+uu_{xxx}=0$$


**Problem 5.** Can the Arnold problem be solved for the α-Proudman–Johnson equations?

Lenells and Misiolek look at the α-connections through α-divergence. Imagine that also in the diffeomorphism group context the α-embeddings produce the same result of the α-divergences, similarly to the finite dimensional case. In such a case the geodesic equation could be the one describing maximum circles on spheres of the $L^{p}$ space thereby solving the Arnold problem for the α-Proudman–Johnson equations.

**Problem 6.** Complete integrability for the α-Proudman–Johnson equations?

If the answer to the previous question is positive, does this help in understanding when one has complete integrability for the α-Proudman–Johnson equations?

**Problem 7.** An Orlicz generalization of α-Proudman–Johnson equations?

If the answer to Problem 5 is positive and we can look at the α-Proudman–Johnson equations as a by product of embedding of densities in the $L^{p}$ spheres, it is natural to ask if using the Orlicz embedding we can get a family of differential equations for which the α-Proudman–Johnson equations is just the particular example associated to $L^{p}$ spaces.

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
