# Peer review of "Lp Unit Spheres and the α-Geometries: Questions and Perspectives"

_entropy, 2020, doi:10.3390/e22121409_

Round 1

Reviewer 1 Report

This paper is an overview including some open problems in classical and quantum information geometry. I think the contents are interesting and this may cause people's attention. The organization of the paper is well, but there are so many places needed to be changed. If the author give the following modifications, I recommend it to be accepted:

P1, ↓line 16, no dot after formula;

P2, ↓line 5, “a n-dimensional ” should be changed into “an n-dimensional”;

P2, ↓line 7, no indentation;

P2, ↓line 9, no indentation;

P3, ↓line 6, change “dual” into “duality”;

P3, ↓line 10, delete dot after formula;

P3, ↓line 18, change “dual” into “duality”;

P3, ↓line 26, delete dot after formula;

P3, ↑line 3, delete dot after formula;

P4, ↑line 3, delete dot after formula;

P5, ↓line 16, change “let's us” into “let us”;

P6, ↓line 3, change “There exist” into “There exists”;

P6, ↑line, change “one see” into “ones see”;

P7, ↓line 17, delete dot after formula;

P7, ↓line 23, change “an hyperplane” into “a hyperplane”;

P7, ↑line 9, change “appear” into “appear”;

P7, ↑line 7, change “appear” into “appear”;

P8, ↑line 12, ↑line 10, ↑line 8, delete dot after formula.

Author Response

All the suggested corrections have been done.

Reviewer 2 Report

See attached pdf

Author Response

1) Basic references about the contents of Section 1-2 have been added.   2) The repetition of the definition of P^1_n has been cancelled. But I maintained the name “probability vectors” in this case because it stresses the finite dimension case.   3) See 1)   4) A comment on the problem of the manifold structure has been placed at the bottom of page 2 (which I hope answer the Reviewer remark).   5) The results are not new and a reference has been added.   6) The  use of the expression “Levi-Civita” for the DUC spaces has been better explained.   7) I’m not sure I understood the point by the referee so I wait for a further elaboration by him, if necessary.   8) A reference for Theorem 5.1 (now Theorem 2) has been added.   9) The word “exacting” has been corrected.

Reviewer 3 Report

this paper does not contain any new results. It's main objective is to provide a list of a few selected open problems in Information geometry whose natural setting is infinite-dimensional. The choice of problems on the list is certainly subjective and influenced by the author's preferences. On the other hand their resolution seems to require a variety of tools from information geometry to functional analysis to geometric mechanics. In my opinion publication of a short note of this type should be of interest to the readers of an interdisciplinary journal like Entropy and may also encourage younger researchers to enter the field.

I recommend publication with the following minor edits:

  1. On p. 4, line 18: "\phi" should be capitalized as "\Phi".
  2. The name "Lenells" is misspelled as "Lenell" in the Abstract, in the title of Sect.11 and several other places in the text and should be corrected.
  3. For completeness of the exposition I suggest that the author provide explicit expressions for the curvatures "c_p" and "Scal_p(\rho)" that appear on pages 5 and 7. This could be done in a separate section providing more geometric background.
  4. Perhaps my only other suggestions to the author would be to combine the last 4 sections under one heading and provide some references on the Proudman-Jonhson eg. Okamoto J. Math Fluid Mech (2009), Wunsch (2011) or Saria and Saxton J. Math Fluid Mech (2013)

Author Response

1) The correction has been made.   2) The name “Lenells” has been corrected everywhere.   3) A section introducing curvature for curves and scalar curvature for manifold has been added   4) The section concerned with Proudman-Johnson equation are now combined in a unique section. The suggested reference for the argument are now in the paper.